Computational modelling reveals the influence of object similarity and proximity on visually guided movements

Patil Mandar Manjunath 1 2
Heinke Dietmar 2
Zhang Fan fan.zhang@xjtlu.edu.cn vanzh89@gmail.com 2 3
1 Department of Neurology, University of California, Davis , Davis , CA , United States of America
2 Department of Psychology, University of Birmingham , Birmingham , United Kingdom
3 School of Robotics, Xi’an Jiaotong-Liverpool University , Suzhou , China
Wang Liang
Electronic publication date: 2025 Feb 25
Publication date: 2025
Volume: 13
Electronic Location ID: e18953
Received 2024 Jun 14; Accepted 2025 Jan 17
Copyright: ©2025 Patil et al.
Copyright year: 2025
Copyright holder: Patil et al.
License: This is an open access article distributed under the terms of the Creative Commons Attribution License, which permits unrestricted use, distribution, reproduction and adaptation in any medium and for any purpose provided that it is properly attributed. For attribution, the original author(s), title, publication source (PeerJ) and either DOI or URL of the article must be cited.
License URL: https://creativecommons.org/licenses/by/4.0/

Keywords: Visual attention, Visual similarity, Visual proximity, Cognitive robotics, Computational neuroscience, Computational modelling, Reach movements, Distraction, Irrelevant feature

Funding: Economic and Social Research Council (ES) ES/T002409/1 XJTLU Research Development Fund (RDF) RDF-23-01-072 This work was supported by a grant from the Economic and Social Research Council (ES) (ES/T002409/1) to DH and the XJTLU Research Development Fund (RDF) RDF-23-01-072 to FZ. The computations were partially performed using the University of Birmingham’s BlueBEAR HPC service. The funders had no role in study design, data collection and analysis, decision to publish, or preparation of the manuscript.

==============================
The paper aims to understand how humans reach for a single target object in multi-object scenes. In a previous empirical study, human subjects were asked to execute reaches to a single target among non-targets (choice reaching task). In the current work, we re-analysed the human data and implemented a neurobiologically-plausible cognitive robotics model (choice reaching with a LEGO arm robot, CoRLEGO) that mimics human reaches in the choice reaching task. The results from the experiment confirmed the commonly made assumption that proximity and similarity between objects (also termed perceptual grouping) affect the quality of the reaches. However, what was novel here was that modelling the reaches also allowed to temporally separate these factors, as the start of the movement was affected by both factors while the reach trajectory was affected only by proximity between target and distractor objects indicating that human information processing of visual stimuli applies these factors in a serial fashion. In particular, our model architecture and the optimised parameter settings suggest that object proximity directly influences the movement onset. In addition, our computational model confirmed this interpretation but also revealed that the relationship between the two factors may be affected by how the participants balanced speed (starting time of the movement) and accuracy of reaching (straightness of reaches). Future research will need to test whether this plausible prediction is correct.

Introduction

The current paper aims to present a neurobiologically-plausible cognitive robotics model of visual attention and action selection termed choice reaching with a LEGO arm robot (CoRLEGO). CoRLEGO primarily focuses on a crucial human capability: visually-guided reaching for a target object in a multi-object environment. The primary aim of this work is not to attain perfection in robotic action planning and control, such as for industrial robots, but rather to mimic the error pattern typically exhibited in human decision-making and relevant behaviours (see Leek, Leonardis & Heinke (2022) for further discussions on this topic). However, it is worth noting that contemporary robotics research assumes that robots with human-like behaviours facilitate human–robot interactions.

Visual multi-object scenes present a strong challenge for the capacity of the human brain as it can only process a limited number of items at once. To deal with this limitation, brain employs selective attention to filter out unwanted information and focus on a subset of objects (but this does not always work perfectly). There is general consensus that selective attention is controlled by top-down and bottom-up mechanisms. Bottom-up processing is a stimulus-driven process in which physically salient items capture attention. Top-down processing, on the other hand, is a goal-driven process in which inputs are evaluated in light of prior knowledge. However, there is a substantial debate on how these two elements affect selective attention (Kinchla & Wolfe, 1979; Egeth & Yantis, 1997). Bottom-up control happens when the features present in the stimuli are unpredictable, whereas top-down control prevails when the emergence of a feature can be predicted (e.g., Muller & Krummenacher, 2006). In addition, there is evidence that perceptual grouping based on Gestalt laws (e.g., spatial proximity and similarity of objects, Todorovic, 2008) affect visual search (e.g., Bacon & Egeth, 1994; Duncan & Humphreys, 1989; Humphreys & Muller, 1993). However, it is also worth noting that some theories suggest that these effects can also be the sole result of saliency (e.g., Wolfe, 2021). Nevertheless, here CoRLEGO employs two factors, saliency and grouping, in its selective attention stage.

These studies typically use button presses as a method to measure participants’ performance. However, Song & Nakayama (2008) and Song & Nakayama (2009) replaced this response mode with reach movements, and subsequently, they called this task the choice reaching task (CRT). In a CRT task, human participants were instructed to detect the search target in a display and touch the target with the finger, so that movement initiation latency (IL) and reach trajectory could be recorded. In turn, the reach trajectory was characterised by its maximum deviation (MD) from a straight line drawn from the starting position to the target. Applied to visual search displays, the CRT task found that many trials trajectories in the CRT task are curved (high MD); they are directed initially towards distractors and then towards the target. Such deviation is especially large if the distractors in the current trial were the target in the previous trial. Song & Nakayama (2008) and Song & Nakayama (2009) explained this effect in the following way: Targets in the previous trial tend to pre-activate the distractor location in current trial (priming effect) and because of the concurrent nature of target selection and movement execution the movements can veer towards the distractors before being corrected leading to curved trajectories. Song and Nakayama termed this effect the “leakage” effect where such visual processes leak into the execution of the reaches. In other words, the CRT reflects the time course of visual selection process. Subsequently, we implemented CoRLEGO allowing us to model this leakage effect and the curved trajectories. Moreover, our studies with CoRLEGO (Strauss & Heinke, 2012; Strauss et al., 2015; Makwana et al., 2023; Zhang et al., 2022) (see also below for details) also indicated that the time course of the target selection first creates a representation of all items in the display, akin to a global representation of the display, which in turn causes the movement to start. Subsequently, the CoRLEGO’s selection mechanism home in on the target resulting into a deviation mainly affected by the local information (i.e., the target). Hence, IL can be seen as being mainly influenced by global characteristic of the display while MD reflects local features such as item positions. Hence, it is conceivable that MD and IL can tap into different processing stages. In fact, below we will show that IL can reflect proximity-based grouping only while IL is affect by proximity-based grouping and similarity grouping.

In the current study, we investigated the effects of irrelevant feature (IF) on guidance of visual search, particularly when the search could be influenced by multiple items displayed simultaneously. The data for the current study were collected in Woodgate’s PhD (2015). The experimental design included tests based on the colour-oddity visual search and CRT, where the subjects search for and reach the odd coloured item (target) under the distraction of other same-coloured items. In this context, the colour being the determinant factor served as the task-relevant feature,while the irrelevant feature was the size dimension. IF was randomised and permitted to vary across all items in the display (refer to examples in Figs. 1A–1C). This randomness in the IF was intended to minimize any potential top-down influence from the predictability of the IF.

Figure 1 Example image input set (A–C) and experimental results (D, E), adapted from Woodgate (2015).

In the set, red is the target color (the middle one in (A & B) and the left one in (C)) whereas the greens are the distractors. The other possible stimuli combinations also considered. Effects of target size and distractor sizes on initiation latency (IL) and maximum deviation (MD) are shown in (D and E), respectively, with small target as a solid line and large target as a dashed line. The abbreviations that describe target or distractors sizes were defined as the following: t = small target, T = large target; dd = small distractors; Dd = mixed distractors; DD = large distractors. Note the displays used in the simulation were shifted to avoid convolutional artefacts (see in Methods for more details).

Previous research has consistently shown that the presence of an irrelevant feature in a search task can significantly influence target selection. The irrelevant features contrast with the other stimuli making them stand out inducing bottom up attention capture. It can be inferred that task-relevant feature with anticipated factors can induce top-down selective attention, but task-irrelevant feature with unexpected factors can cause bottom-up attention capture (Bacon & Egeth, 1994; Woodgate, 2015; Theeuwes, Atchley & Kramer, 2000). Additionally, Woodgate (2015) assumed that grouping two large distractors (DD) via proximity grouping would facilitate the search. The search performance would be disrupted if distractors were dissimilar (Dd) because they cannot be grouped by similarity. Moreover, the negative impact is stronger when the large target groups with the large distractor via proximity (TDd). In Woodgate’s study results, as depicted in the Fig. 1 bottom panel, the reduction in IL with the increase in distractor sizes suggests that the proximity grouping effect influences the IL. A similar effect is observed in MD as a significant decrease from dd to DD exists. But a negative effect on search is an increase in MD value is seen when the two distractors cannot be distinguished through similarity, and the large target and large distractor group together through proximity. Based on this effect, Woodgate suggested that similarity grouping only influences the late selection process, affecting only the MD. However, proximity grouping is responsible for influencing both IL and MD.

The current work intends to computationally model the above-mentioned experimental evidence using a neurobiologically inspired robotics model called CoRLEGO (choice reaching with a LEGO arm robot) (Strauss & Heinke, 2012; Strauss et al., 2015) and its extension SH-CoR (selection history in choice of reaching) (Makwana et al., 2023), to model how current reach target selection can be best explained by the mechanism postulating the dynamic interplay between distractor inhibition and target facilitation in selection history effects. Here we adapt the model to accommodate IF conditions and image inputs by introducing architectural changes which is explained in the following sections. Finally the model is extensively parameterised through trial and error procedure to match the experimental evidence. In literature, models have already aimed at understanding attention selection effects by employing drift diffusion models (Tseng et al., 2014; Burnham, 2018; Allenmark et al., 2021) and the TVA-model (Ásgeirsson, Kristjánsson & Bundesen, 2015). While these models have provided valuable insight in general, how action selection was modulated with evolving continuous goal-directed movements have not been addressed as in SH-CoR. Moreover, SH-CoR uses competitive selection mechanisms commonly used to model visual selective attention (Heinke & Humphreys, 2003; Heinke & Humphreys, 2005; Narbutas et al., 2017; Abadi et al., 2019), which allows SH-CoR to capture how both attention and action selection mechanisms influence reach movements (i.e., leakage effect). Taken together, the combined mechanisms makes SH-CoR the most suited model to understand the behavioral data in Woodgate’s experiments.

Methods

Cognitive robotics approach—an extension of CoRLEGO

CoRLEGO was initially designed by Strauss & Heinke (2012) to simulate the theory of information leakage from the visual cortex to the motor cortex in an odd colour choice reaching task (Song & Nakayama, 2008). It was later summarized and discussed how the different stages of the model relate to the brain areas in Strauss et al. (2015). Subsequently, CoRLEGO was extended as SH-CoR to model the dynamic interplay between target and distractor features in selection history effects (Zhang et al., 2022; Makwana et al., 2023), further expanding its applicability to cognitive processes involving selection history. This model consists of two central processes—the target selection stage and the motor stage—which are topologically linked and operate in parallel. For a detailed depiction of these stages, refer to Figure 2 in Strauss & Heinke (2012).

The target selection stage employs SAIM (Selective Attention for Identification Model) to identify the odd colour target from the extracted colour maps of the display. Here the node with the highest activation is chosen as the output (Heinke & Backhaus, 2011). The motor control stage encodes the information from the target selection stage as an activation blob using the dynamic neural field (DNF) theory (Erlhagen & Schoner, 2002). This activation blob encodes the velocity motor parameter that can be translated into the arm movement of the robot using inverse kinematics. The model is a closed-loop robotic system that takes the visual scene as input and generates arm movement based on the competitive processes at different stages (Strauss & Heinke, 2012; Strauss et al., 2015; Makwana et al., 2023; Zhang et al., 2022). The Target Selection and Motor stage modules represent “the main model” and operationalise the features of Song & Nakayama’s theory (2008) and resembling the development in SH-CoR (Makwana et al., 2023).

Implementation of the visual processing module

The original implementation involved modelling a simple choice-reaching task in which an odd colour target had to be identified from an experimental stimulus. Image processing were employed to detect the red or green colours (with more colours in the display allowed by SH-CoR); identify the odd target and create competitive colour feature maps. The processed information was further fed and computed in the other modules (Strauss & Heinke, 2012; Strauss et al., 2015).

But to account for the size IFs in the extension, the extraction of the size-related salience of the items along with the target (odd-colour) identification was required. Saliency levels were assigned to each item based on both colour and size. The saliency levels represent a saliency score given to an area of interest that reflects its visual prominence (Clarke, Dziemianko & Keller, 2014), thereby improves the interpretability of the model.

For a given experimental stimulus, the visual processing module pre-processes the stimulus, extracts the items in the display and compares them to find the configuration type. The algorithmic schema (Fig. 2) of the visual processing module is the following: The experimental stimulus (coloured image input) is resized and converted into a gray-scale image before being converted to black and white. This transformation aids in identifying the image’s connected components (Jeong, 2019). The connected components represent the various items present in the experimental stimulus. The colours of each item are scanned to identify the odd colour, which allows the items to be classified as a target or a distractor. Furthermore, the sizes of the items are compared to determine whether they are large or small. The module assigns a predefined saliency level based on the item configuration detected. The saliency levels are designed such that similar items have same saliency levels, while targets will have higher levels than the distractors, indicating their greater visual prominence. The predefined levels are arbitrary and need extensive tuning to obtain desired model performance. This tuning process involves testing several combinations to identify the most effective settings. These saliency levels are further used by motor modules for movement production.

Figure 2 The algorithmic schema of the visual processing module.

The visual processing module is the first module that received image input and connects to the rest of the modules in the mode. Input is the image of size (30 × 30). Output is the location information of each item for spatial location map, saliency level predefined per item for competition, and the proximity information as a scaling factor for movement onset.

Mimic human-like reach

The original target colour map is replaced in the target selection stage (see Fig. 3) by the combination of saliency competition nodes and spatial location maps. It selects a target in the target location map based on the saliency levels extracted by the visual processing module. The modules in motor stage are kept the same as in the original model, with changes to the parameters of the DNF elements for different tasks and behaviors. Successful completion of a simulation trial is characterised by a curved reaching trajectory selecting the target. This was achieved from a competitive selection because of the perceptual grouping effects of irrelevant features (Song & Nakayama, 2008; Woodgate, 2015). Also, the velocity profile should resemble a bell-shaped curve signifying human-like behaviour during reaches in a goal-directed task (Flash & Hogan, 1985). The new architecture depicts the addition of spatial location maps and saliency competition nodes to the target selection stage of the model (see Fig. 3).

Figure 3 Extended model architecture with the saliency competition nodes and spatial location maps (created with BioRender.com).

Note that the red/green colors used in the display image input (see Fig. 1) were adjusted for demonstration purpose, i.e., to make them accessible to those with color blindness.

Modelling target selection

Three competition nodes were created, one for each item in the display, instead of one for each colour as in previous models. The associated saliency levels are fed into these nodes. The dynamic neural field (DNF) theory governs the behaviour of these nodes (Erlhagen & Schoner, 2002). However, a new implementation of a recurrent on-centre, off-surround network (i.e., a RNN) replaced the original implementation of the single neuronal dynamics (Grossberg, 1973; Amari, 1977). It incorporates coefficients for local excitation and global inhibition between nodes and allows competing behaviour. The rate of change of the node activation, ui ˙, was implemented as follows: (1) τ⋅ui ˙=−A⋅ui+B⋅fui+C⋅Ii+D⋅∑fui+hi+ϵ,

(2) fui=1/1+eβui−u0

Here for given node i, ui represents the activation, h stands for the resting level, A is a constant equals to 1 in the current work, B is the self-excitation level of the node, the sigmoid function f(ui) is the output of the current node, Ii is the input to the node (in this case, the saliency level), C is the coefficient for local active nodes, D is the coefficient for the global active nodes, τ is the time factor of how fast the node adapt to the change in the parameters and ϵ is the random gaussian noise for robustness. The sign and magnitude of the coefficients, C and D, determine the corresponding excitation/inhibition effects from the neighbouring nodes. The tuned coefficients (see Table 1) employ a winner-takes-all (WTA) mechanism where a node with the highest activation is selected while competing with other nodes. The activations of the nodes depend proportionately on the saliency level assigned. Implementing the modified node dynamics as above will imply a lower inhibition effect between the nodes when the difference of their activation is less and vice versa. The reduced inhibition effect will lead to a longer suppression time of the other nodes whilst the winner node is selected. As a result, the saliency levels are carefully selected after proper tuning. As in Woodgate’s experimental results, the chosen saliency levels should be such that the larger distractors must have a less inhibitory impact on the target than the smaller ones. As a result, the saliency levels are constructed in the descending order: small target > large target > small distractor > large distractor

Table 1 Parameters of saliency competition node implemented (Eqs. (1) & (2)).

τ	h	β	A	B	C	D	ϵ	
20	−1	1	1	7	0.7	−4	0.005	

Spatial location maps are created to represent the spatial features of the items in the display. These maps have Gaussian inputs centred at the location (centroid) of each item present in the stimulus. The output from each of the saliency competition nodes is multiplied with the corresponding spatial location maps, and these three maps are summed together to create a single map of the size of the display item (e.g., the black star in Fig. 3). This resulting summation is input to the target location map. A topological connection between the saliency competition and the target location maps exists. The representation of the items on the target location map appears even before a single target is selected. Once the competition in the saliency competition nodes is over, the units at the target location will only be activated in the target location map. These parallel representations cause the curved trajectories consistent with Song and Nakayama’s theory (2008), and the competition in the saliency competition nodes are consistent with grouping interactions, as in Woodgate (2015). The output from the target location map is relayed into the motor stage of the model.

Motor planning and movement production

The motor stage encodes the location information from target selection modules into motor parameters using activation blobs governed by the DNF framework (Erlhagen & Schoner, 2002; Schoner, Spencer & Group, 2016). The elements of the motor stage are retained the same as in SH-CoR, with only a few modifications to their DNF parameters (see Makwana et al. (2023) for more details). The motor stage comprises three maps: the hand map Hloc (a 2D Gaussian), the hand-target difference map (Displacement Representation D map), and the velocity map (V map). The displacement representation was created through a convolution of hand map with the target location, i.e., output of target selection (Tloc), i.e., via a CNN: (3) Dx¯,t= ∫Hlocy¯−x¯,t⋅fTlocx¯dx¯

where x and y represent the hand position as a function of time in each simulated step, we update the hand position based on the velocity in a close loop control. The D map forms the input of V map plus another 2D Gaussian to initiate activations: (4) IVel1x¯,t=Dx¯,t+Amp⋅ex¯2/2σ2.

The velocity representation uses a two-layer DNF namely layer Vel1 and Vel2 to generate and stabilise the blob-shape activation: (5) τ⋅x ˙iVel1=−xiVel1+IVel1+exclocxiVel1+inhxiVel1+hi+ϵ,

(6) τ⋅x ˙iVel2=−xiVel2+exclocxiVel2+inhxiVel1+hi+ϵ,

(7) exclocx¯= ∫wexcx¯−x¯′⋅fx¯′dx¯′

(8) inhx¯= ∫winhx¯−x¯′⋅fx¯′dx¯′+ginh⋅ ∫fx′dx′

in which the kernel w with strength c and width parameter σ as defined with: (9) wx¯=cσ2π⋅exp−|x¯|22σ2.

Here, the movement direction information is encoded into the velocity parameter. The velocity parameter is then converted into end effector joint velocities using inverse kinematics. Also, the information is fed back into the model to update the current hand position in the hand map. Random noise is added at different stages: the saliency competition nodes, the velocity map and the hand target difference map, to model the variations in the human trials.

Parameter tuning

The model is tuned in a trial and error fashion to adjust the parameters to get the desired activations at the different topological levels of the model (Table 2). As discussed in previous sections, the designed saliency levels are arbitrary; tuned appropriately along with the model’s DNF element parameters. The saliency levels are set up so that the target has the highest saliency level and gets chosen. The current saliency levels for various item types are shown in Table 3.

Table 2 Parameters of other DNF elements which follow original neuronal dynamics implementation.

Element name	τ	h	β	g inh	c exc	σ exc	c inc	σ inc	
Target location map	20	−1	4	
Hand-target difference map	30	−5	4	
Velocity location map	30	−0.2	4	
TL to TL				0	3	7.5	1	15	
D to D				−0.01	3	5.5	0	0	
D to V				−0.005	3	5.5	1	10	
q to D				0	0.005	1	0	0	
q to V				0	0.005	1	0	0	

Table 3 Saliency levels assigned for different item types present in the stimulus.

Item type	Saliency level	
Large target (T)	4.8	
Small target (t)	5.0	
Large distractor (D)	4.4	
Small distractor (d)	4.6	

For example, the ddT configuration: a small distractor on the left, another small distractor in the middle, and a large target on the right will be assigned saliency levels, 4.6, 4.6 and 4.8, respectively. For the dissimilar distractor condition, both the possible positions of the distractors are considered (large distractor can be either left or right). The stimuli are permutated and given to the model in line with the literature to avoid top-down influence from irrelevant feature predictability (Bacon & Egeth, 1994; Woodgate, 2015).

After picking set of parameters, the model was validated for 25 sets, each with 24 stimuli images (30 × 30 pixels) corresponding to different possible item combinations (as shown in Fig. 1), totalling 600 trials. A noise of magnitude 0.005 at the saliency competition nodes, the hand-target difference map, and the velocity map was added to model the variations in the human trial. At the end of each trial, a break time was introduced to clear the residual activations in the maps so that they won’t influence the successive trials. Based on the saliency levels assigned to items, competition occurs at different model stages. The model moves the end effector to create a hand trajectory when there is sufficient activation in the velocity map. But a threshold of 0.5 is set in the velocity map to avoid random noise from moving the hand. At the end of each trial, the hand velocity and the hand position of the end effector are registered.

Data analysis and model evaluation

The IL and MD can be calculated using the hand velocity and the hand position of the end effector. IL is calculated as the time steps it takes the end effector in a trial to reach 10% of the model’s maximum velocity of the hand across all trials. The hand velocity of each trial is fitted to a polynomial of degree 10 to improve the resolution of the IL. This 10% is an arbitrary value representing the minimal activation required to initiate the hand movement. MD can be calculated as the largest perpendicular distance between data points on the trajectory and the straight line drawn from the start of the hand position to the target. Using IL and MD, a qualitative assessment can be made of whether the modelling results were qualitatively similar to human performances. For example, CoRLEGO results must demonstrate a significant decrease in IL when the distractor size increases, which indicates the facilitation of search. The model results can be normalised and compared by analysing how closely they fit the experimental results. In computational modelling experiments like CoRLEGO, qualitative analysis is more appropriate than quantitative analysis (see Heinke (2009) for further discussion).

Results

Examples of the trajectories for six trials are shown in Fig. 4A, and they show that the reach trajectories (blue line) select the correct target. Figure 4B depicts the example of a trial’s hand velocity in blue and its corresponding fitted curve in orange, showing the human-like bell-shape velocity profile. The blue horizontal dashed line indicates the maximum velocity (around 0.053 in Fig. 4B). The velocity threshold θ for determining the timing of movement initiation was set at one tenth of the maximum velocity (0.0053). Here, the red vertical line indicates the resulting IL (15 steps).

Figure 4 Left: the reach trajectories of six randomised trials are indicated by the blue line. Right: the bell-shaped hand-velocity profile of one trial’s record (blue) and fitted polynomial curve (orange).

The IL is calculated at 10% of the maximum velocity across all trials (threshold).

In first study, we generated 600 trials without the scaling factor. Figure 5 shows the results for IL and MD averaged across trials for each target location and display configuration separately. Importantly, no IL effect was observed in all configurations, which is inconsistent with the experimental findings. In contrast, for MD it is consistent with human data with the exception of target in the middle condition, where only very small MD effects were observed. This was mainly due to the model producing reaches in two-dimensional space, where the effects of the distractors on either side of the target on the movement cancelled each other out. In the next analysis, we only consider the target on the left and the target on the right conditions.

Figure 5 IL and MD plots for the trials separated for each target location for the simulation without the movement onset scaling factor.

Failure in simulate IL without the movement onset scaling factor.

In the second study, we generated results testing the scaling factor. A scaling factor of 0.35 for dd, 0.2 for Dd, and 0 for DD were implemented, and the observed results are shown in Fig. 6. It shows that IL successfully yielded the pattern of human results (dd > Dd > DD). The overall pattern of the MD in experimental data was relativly similar to the human results (Dd > dd, Dd > DD), but we also observed a higher MD in the DD condition relative to dd condition which is contradictory to human results. We further optimised the scaling factor and found that the results fitted to human data the best with 0.125 for dd, 0.1 for Dd, and 0.075 for DD. As shown in Fig. 7, both IL and MD exhibit patterns analogous to those observed in human studies.

Figure 6 IL and MD plots for the trials separated for each target location for the simulation with movement onset scaling factor values, dd: 0.35, Dd: 0.2 and DD: 0.

Note that an unwanted increase in deviations from the condition dd to DD was observed. See the left panels in Fig. S1 for the IL and MD plots of the target-in-the-middle trials.

Figure 7 IL and MD plots for the trials separated for each target location for the simulation with movement onset scaling factor values, dd: 0.125, Dd: 0.1 and DD: 0.075.

See the right panels in Fig. S1 for the IL and MD plots of the target-in-the-middle trials.

Discussion

Pre-attentive visual processing and saliency detection

In our work, the visual processing module executes the brain’s pre-attentive processing of visual stimuli which fall into two main functions as theorised by Muller & Krummenacher (2006). First is the extraction of basic attributes. Attributes such as colour, motion direction, and velocity correspond to features recognised and registered pre-attentively (Wolfe, 1998). These basic features are the stimulus properties responsible for activating single cells in the early visual areas. The initial image processing techniques employed in the visual processing module correspond to the first function. The second function of pre-attentive processing is to direct the focus of attention to the essential information derived from the first function’s basic attributes. In our model, the assignment of saliency levels corresponds to this function.

Model v.s. Human

IL and MD averaged across all trials are normalised and compared (Fig. 8). The sum of the square root of error between the model results and human data (Fig. 2) is 0.012 for IL and 0.0095 for MD, indicating a very close fit. The saliency competition nodes that have been implemented can generate a reaching behaviour that selects the correct target based on the assigned saliency. The average velocity’s bell shape represents human-like behaviour (Flash & Hogan, 1985) unlike that of the industrial robots. The assigned saliency levels of items can create curved trajectories indicating the capability of the distractors to influence the reach that mimics human behaviour. The IL effects observed in human trials cannot be replicated solely by saliency mapping (see examples in Fig. 5).

Figure 8 The normalised average values of the datapoints of IL and MD from the experimental data and the model simulation data overlaid over each other, with movement onset scaling factor values, dd: 0.125, Dd: 0.1 and DD: 0.075.

A scaling factor for movement onset based on the proximity information of the distractors accounted for the IL fitting, that novelly connected the visual processing and movement control, representing the strength of the coupling effect. Having the right set of values for the movement onset scaling factor is necessary. For example, a factor of 0.35 for dd, 0.2 for Dd, and 0 for DD would yield a pattern like the human IL results. But then there is an increase in deviation observed from the condition dd to DD (Fig. 6). The results violate Woodgate’s claim that the increase in the size of the distractors allows for easy target identification. This is an example of a simulation where the IL results are good, but MD is not. After extensive parameterisation, we found the optimised scaling factor of 0.125 for dd, 0.1 for Dd and 0.075 for DD (Fig. 7).

Furthermore, when the target is in the middle, the deviations observed are smaller than those observed at the other two locations. The activations from the distractors on either side average out to almost the middle position, resulting in no pull to cause the curved trajectories. Therefore, the results for the target-in-the-middle trials are not shown in Figs. 6 and 7 but are instead reported in Fig. S1. Previous research has found a similar average activation effect in the motor cortex’s encoding of movement direction by the neuronal population (Georgopoulos et al., 1982). Woodgate (2015) did not explicitly analyse the results of the middle target’s maximum deviation. The observed results may imply that when distractors are on either side of the target, they have little effect on search performance because they cannot be easily grouped to induce any perceptual grouping effects.

General discussion

Woodgate (2015) investigated the effect of irrelevant feature (IF) on search performance when it coincided with multiple items on display using CRT experiments on humans. The aim was to dissociate the two grouping effects, proximity, and similarity grouping, using the IL and MD measures of the CRT experiment. When the distractors could not be grouped via similarity, the search performance degraded. The negative effect on the search was especially strong when the large target grouped with the large distractor via proximity, and the two distractors could not be grouped via similarity. Woodgate suggested that the proximity grouping affected both the brain’s early and late selection stages. In contrast, the similarity grouping only affected the late selection process.

Note that previous research has shown that the proximity and similarity grouping effects rely on temporally distinct processes. In a recent work, Johansson & Ulrich (2024) systematically investigated the interplay between proximity groups and similarity groups in two experiments and found serial processing of the groups with a self-terminating rule for processing. Trick & Enns (1997) found that clustering and shape formation in grouping contributes differently in vision. In their global level letter discriminative tasks (H or E comprised of local circles), Han and colleagues (1999) discovered that proximity grouping of local circles promoted shorter discrimination times rather than similarity grouping. It is further supported by an event-related brain potential (ERP) recording study of the same task, which shows that proximity grouping is associated with early activity in the medial occipital cortex and late activity in the occipito-parietal cortex, whereas similarity grouping only influences late activity in the occipitotemporal region (Han et al., 2001; Han, Ding & Song, 2002). Later, among other factors in grouping, Sasaki (2007) found that V1 is involved in grouping by proximity, while grouping by similarity relies on higher visual areas.

In this paper, we implemented a neurobiologically plausible cognitive robotics model to mimic human reach in Woodgate (2015). Our model can replicate the human performance closely for both IL and MD. The movement onset scaling factor based on the proximity information of the distractors determines the ease at which the participants can begin their movement. Furthermore, the movement based scaling mechanism for induced delay can reflect how well the participants set themselves up for the task. It suggests that participants tried to make ’a happy compromise’, i.e., the trade-off between speed and accuracy (SAT, the speed-accuracy-trade-off), based on the visual stimuli to select the right target as soon as possible, i.e., early stage target selection process is involved and coupled.

Our model had object saliency levels based on distractors’ similarity, and the movement onset scaling factors based on proximity between target and distractors. By optimising these parameters and fitting to IL and MD in human data, we found that it worked the best when similarity (between distractors only) contributed to saliency and their competitions in target selection, while object proximity was directly linked to movement production, and only with tunning the scaling factor the IL could be fitted well. In other words, our model showed that proximity information may not be involved in target selection process, including the process of extracting saliency and their competitions. Therefore, whether proximity and similarity group contribute to early or late stage of selection process in the brain, are in doubt. However, the success of our model fitting confirms that the presence of IF disrupts search performance via the similarity and proximity grouping effects.

Limitation and Future Work

One limitation is that the current experimental setup only had two distractors in the input displays. The results cannot be generalised to visual scenes with more distractors in a complex scene. A higher number of items would promote multiple groups and cause behaviours, unlike the ones analysed here. Further studies could simulate the effect of having more than two distractors by accommodating them into the current architecture.

The computational modelling and cognitive robotics frameworks could aid in a better understanding of the underlying brain mechanisms of the above-mentioned human behaviours for questions that cannot be addressed experimentally. The cognitive robotics approach presented in this paper can be used in future research to simulate results from other behavioural studies testing human decision-making and perception and is not limited to visual attention capture experiments using CRT tasks. A manipulation of SAT could also be implemented to systematically test the trade-off effects between the speed and accuracy of the participants in the selection and action planning.

Supplemental Information

Supplemental Information 1 Simulation code (Matlab) and Toolbox

Supplemental Information 2 IL (top) and MD (bottom) of the target-in-the-middle trials for the simulation with movement onset scaling factor values as in Fig.6 (left) and Fig.7 (right)

Each data point represents the average performance across trials for a given item combination (dd, Dd, DD) under different size conditions (small: blue squares, large: red stars). The upper panels show IL (interaction latency) in terms of time (in steps), while the lower panels depict MD (movement distance) in terms of distance (in pixels). The solid blue line corresponds to the small target condition, and the dashed red line represents the large target condition.

Additional Information and Declarations

Competing Interests

Author Contributions

Data Availability

The authors declare there are no competing interests.

Mandar Manjunath Patil conceived and designed the experiments, performed the experiments, analyzed the data, prepared figures and/or tables, authored or reviewed drafts of the article, and approved the final draft.

Dietmar Heinke conceived and designed the experiments, authored or reviewed drafts of the article, and approved the final draft.

Fan Zhang conceived and designed the experiments, analyzed the data, prepared figures and/or tables, authored or reviewed drafts of the article, and approved the final draft.

The following information was supplied regarding data availability:

The data is available at OSF: Zhang, Fan, Dietmar Heinke, and Mandar Patil. 2025. “Computational Modelling Reveals the Influence of Object Similarity and Proximity on Visually Guided Movements.” OSF. January 22. osf.io/xzqdy.

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
