# Peer review of "Computational modelling reveals the influence of object similarity and proximity on visually guided movements"

_PeerJ, doi:10.7717/peerj.18953_

## Round 0.1 · original submission · Major Revisions

· Academic Editor

Major Revisions

Please respond to the reviewers’ comments and concerns carefully. We look forward to receiving your revised manuscript.

Reviewer 1 ·

Basic reporting

The current study implemented a so-called CoRLEGO robotics model to mimic human action - reaching for a target among multiple objects. The alleged novelty is that perceptual grouping like proximity and similarity influence the reach trajectory and a key scaling factor is necessarily implemented in the model for this purpose. Although the topic is interesting, I have several concerns that prevent me to support the publication of the manuscript.
One major concern is that I don’t quite see the purpose for mimicking the exact properties of human action. In my view, we build models to mimic human cognitive process for its high intuitiveness/creativity, efficiency and low requirement on energy consumption. But generally speaking, human actions are not accurate, so what’s the benefit for mimic human action? On the other hand, building a robotic model for action does not seem to advance our understanding on human cognitive processing mechanism either. The authors may want to clarify this.

Experimental design

1. The interpretation for the same size large/small stimuli as similarity and proximity seems to me an over generalization in the Gestalt grouping theory. I’m not saying that stimuli with the same size can’t have similarity grouping, nor that stimuli close to each other can’t have proximity grouping, but the experimental setting in this particular study was mainly related to separation (distance) between stimuli and size consistency, I don’t see grouping necessarily involved.
2. The authors claimed that initiation latency (IL) reflects early global processing, while maximum deviation (MD) reflects late selection process, why is so? Why global processing is an early stage of processing? And MD to me is most likely to reflect inaccuracy in executing the action, which has nothing to do with selection process.

Validity of the findings

1. I’m not an expert in computational/robotic models, but it seems to me that all this scaling factor does is to scale the info/signal based on size consistency. Can it be incorporated in the saliency map? After all, size consistency may also reflect saliency. And how is it mathematically defined and implemented?
2. The authors may want to discuss and compare other related models, to readers a better .

Additional comments

1.The font size in figure 5 was too small to see. The authors should provide more detailed descriptions for Figures 7 -9.
2. I would suggest more extensive copy editing. There were a number of syntactical errors, e.g, abstract, ‘however, novel was that…’.

·

Basic reporting

Reviewer comment: Abbreviations of the form P7L218 refer to page number and line number (i.e., page 7, line 218)

P1L21: “indicating that human info processing of visual stimuli applies these factors in a serial fashion”
Should be perhaps:
“indicating that human information processing of visual stimuli applies these factors in a serial fashion”

P1L38: “To deal with this limitation brain employs selective attention […]”
should be perhaps:
“To deal with this limitation, the brain employs selective attention […]”

P2L59: “attend to target” should be “attend to a target”

P6 Figure 5: A text element in the box-and-arrow model misspells the word difference as “Differennce”

P7L217: “In which the kernel w with strenth c and width parameter […]”
this sentence should have italicized w and ‘strenth’ should be ‘strength’.

P8L243: “The initiation latency and maximum deviation can be calculated […]”
The manuscript is inconsistent in that it sometimes uses abbreviations and sometimes it does not. Make sure that the abbreviations are introduced in parenthesis the first time they occur, e.g.:
“The initiation latency (IL) and maximum deviation (MD) can be calculated […]”
Also use them consistently or not at all.

P9L279-282: “But the maximum deviation, even though the overall pattern looks like the human results (MD values it was Dd>dd, Dd>DD). There is an increase in deviation observed between the conditions dd and DD. The results are opposed to the claim that increase in the size of the distractors allows for easy target identification as the MD is greater for DD than dd when the target is on left/right.”
The phrasing here is confusing and the paragraph could use some rewriting. Also, are you refering to a claim or an empirical finding here? I understood it as the later. In that case, the following reads clearer to me:
“Although the overall pattern of MDs in the simulated data resembled human data (MD values Dd>dd and Dd>DD), we also observed larger MDs in condition DD relative to condition dd. Model predictions were therefore at odds with the findings of McSomeone (xxxx) who observed smaller MDs in condition DD in choice-reaching task with human observers.”

P12L334 "Note that previous research […]”
Some other important references supporting serial processing of proximity groups and similarity groups are:
Trick, L. M., & Enns, J. T. (1997). Clusters precede shapes in perceptual organization. Psychological Science, 8 (2), 124–129. https://doi.org/10.1111/j.1467-9280.1997.tb00694.x
Sasaki, Y. (2007). Processing local signals into global patterns. Current Opinion in Neurobiology, 17 (2), 132–139. https://doi.org/10.1016/j.conb.2007.03.003
I recently wrote a paper on this topic which might be of relevance to the authors:
Johansson, R. C. G. & Ulrich, R. (2024). Serial processing of proximity groups and similarity groups. Attention, Perception, & Psychophysics. https://doi.org/10.3758/s13414-024-02861-2

P13L364-365 “Also, proximity grouping is prominent over similarity grouping Duncan and Humphreys (1989).”
This should be a parencite not a textcite, i.e. “[…] over similarity grouping (Duncan & Humphreys, 1989)”

Experimental design

The chief aim of this paper was to computationally model human behavioral data collected in a previous experiment. This previous research investigated the effects of perceptual grouping factors on visually guided movement in a choice-reaching task. In other words, the methodological approach of this research was primarily computational and not empirically oriented. I shall therefore summarize and review the computational approach of the paper in this section.

In short, the authors develop a computational robotics model which builds an extends on the CoRLEGO approach of Strauss & Heinke (2012). Like many potential readers of this paper, I am not previously familiar with CoRLEGO. However, both quantitative and qualitative aspects of this modelling framework are described in text, and key equations are provided. In my opinion, enough information is provided to guide the reader through this dense topic.

The championed model contains two computational components: a visual processing module which receives image inputs that are similar to the visual stimuli used in choice reaching tasks. Each image is scanned for two types of elements: targets and non-targets. Elements are identified on the basis of color (target dimension) and size (distractor dimension) using the Selective Attention for Identification algorithm developed by the second author. Color and size features are then converted to a saliency value. The most salient locations of visual space guides reaching behavior by means of a competitive, winner-takes-all dynamic between processing nodes in the second model stage (motor module). Crucially, the nodes of the motor module (representing the elements of the visual array) are connected horizontally in such a manner that they reproduce the effects of perceptual grouping by proximity and similarity on movement trajectories and velocities.

In general, this modeling approach appears to me to be a a well-suited methodology for investigating the relation between configural visual processing (grouping) and visually guided action. I also welcome that the paper is anchored in formal mathematical modeling. The attempt to emulate human perceptual grouping effects on behavior in a robotic agent is also a nice, novel feature.

Validity of the findings

Simulation results and parameter estimates are clearly reported in figures and tables. Code and data are provided.

The authors underscore that a qualitative analysis of simulation results is an important tool in evaluating model fit. This potentially opens up the door for some ambiguity in interpreting results, but I think it is balanced well in this paper. Results are not overstated or misrepresented.

Both reaching trajectories and the distribution of movement initation latencies are well-reproduced by the computational model, suggesting that the advanced framework has some merits.

Figure 10 in particular provides good support for the final model. In my opinion, it would be nice to have 95% error bars added to the human datapoints here.

Additional comments

P9L273-274 “But the initiation latency is measured the same across all the configurations which is contrary to the human results.” Perhaps you could again explain in a sentence how the pattern of initiation latencies looks for human observers? I know this is explained in the next paragraph, but readers generally have poor short-term memory for these kinds of things and you should repeat it before you go on to the next figure.


P10 Figures 7 and 8: It seems the blue and orange datapoints overlap to a very large extent here, in particular, in the upper rows of both figures. Can you make use of blue circles and orange triangles or smth similar to make this more immediately evident to the reader? Also, striped connecting line segments might help make all datapoints visible.


On a final note, I am quite sure the authors could think of shorter and more informative title. Something closer to 10-15 words than the current 22 might make the paper more memorable to readers. For example:
A computational robotics account of perceptual grouping factors in visually guided human movement
Or:
Differential impact of similarity and proximity on visually guided movement: A computational analysis of human performance
But I do not want to overstep here, so they decide if thinking of a shorter title is worthwhile.

---

## Round 0.2 · Minor Revisions

· Academic Editor

Minor Revisions

Please address the reviewer's comments and submit your revised manuscript with a point-by-point response. Thanks. We look forward to receiving your revised manuscript.

Reviewer 1 ·

Basic reporting

In the present version of the manuscript, the authors have made considerable efforts to address reviewer comments and incorporate valuable edits. I believe that the authors have addressed my concerns.
I do have one more suggestion: in the current manuscript, although I understand the reason why the authors have removed the results for target in the middle condition, these results perhaps should be reported in supplemental materials.

Experimental design

No comment

Validity of the findings

No comment

·

Basic reporting

No comment.

Experimental design

No comment.

Validity of the findings

No comment.

Additional comments

All of my suggestions for improvement have been tended to by the authors. I recommend accepting the paper.

---

## Round 0.3 · accepted · Accept

· Academic Editor

Accept

The authors have revised the manuscript based on the reviewer's comments, and it can be accepted now.